# Classification of Chemotherapy-Related Subjective Cognitive Complaints in Breast Cancer Using Brain Functional Connectivity and Activity: A Machine Learning Analysis

**DOI:** 10.3390/jcm11082267

**Published:** 2022-04-18

**Authors:** Lei Wang, Yanyan Zhu, Lin Wu, Ying Zhuang, Jinsheng Zeng, Fuqing Zhou

**Affiliations:** 1Department of Radiology, The First Affiliated Hospital of Nanchang University, Yongwaizheng St. 17, Nanchang 330006, China; dr_leiwang@126.com (L.W.); zyanyan06@126.com (Y.Z.); ndyfy04061@ncu.edu.cn (L.W.); 2Neuroradiology Laboratory, Jiangxi Province Medical Imaging Research Institute, Yongwaizheng St. 17, Nanchang 330006, China; 3Department of Oncology, Jiangxi Hospital of Integrated Traditional Chinese and Western Medicine, Bayi Ave. 90, Nanchang 330003, China; 4Department of General Surgery, The First Affiliated Hospital of Nanchang University, Yongwaizheng St. 17, Nanchang 330006, China; zengjinsheng0606@163.com

**Keywords:** resting-state functional magnetic resonance imaging, machine learning, breast cancer, chemotherapy-related cognitive impairment, subjective cognitive complaints

## Abstract

The aim of this study was combining multi-level resting-state functional magnetic resonance imaging (rs-fMRI) features with machine learning method to distinguish breast cancer patients with chemotherapy-related subjective cognitive complaints (SCC) from non-chemotherapy (BC) and healthy controls (HC). Forty subjects in SCC group, forty-nine in BC group and thirty-four in HC group were recruited and underwent rs-fMRI scanning. Based on the anatomical automatic labeling brain atlas, the functional metrics of all subjects included functional connectivity, amplitude of low frequency fluctuation and fractional amplitude of low frequency fluctuation, regional homogeneity, voxel-mirrored homotopic connectivity and degree centrality were calculated and extracted as features set. Then, the rs-fMRI features were selected by two-sample t-test, removing variables with a high pairwise correlation and least absolute shrinkage and selection operator regression. Finally, the support vector machine models were built for classification (SCC vs. BC, SCC vs. HC). Thirty-eight features (SCC vs. BC) and seventeen features (SCC vs. HC) were selected separately, and the accuracy of the models were 82.0% and 91.9%, respectively. These findings demonstrated a valid machine learning approach that effectively distinguished breast cancer patients with chemotherapy-related SCC from non-chemotherapy and healthy controls, providing potential neuroimaging evidence for early diagnosis and clinical intervention of chemotherapy-related SCC.

## 1. Introduction

Breast cancer is the most common malignant disease that threatens women all over the world [1], and chemotherapy is one of the most important treatments. More than 50% of breast cancer patients have cognitive impairment in many aspects, including memory, execution, attention and reaction speed, during or after chemotherapy (also known as “chemo-brain”), and more than half of them are mainly manifested as subjective cognitive impairment (SCC) [2,3], which seriously affects the ability to work and socialize [4]. With the continuous development of medical technology, the number of breast cancer survivors is increasing, and the social and economic burden brought by chemotherapy-related cognitive impairment (CRCI) is becoming heavier. Therefore, early identification of the occurrence of CRCI is crucial for early clinical intervention and prognosis of patients.

Several neuroimaging studies have shown the reduced volume and density of brain gray matter and the destroyed structural integrity of white matter in CRCI [5,6,7]. Additionally, the resting-state functional magnetic resonance imaging (rs-fMRI) studies had also confirmed that there were extensive abnormalities of local functional activities and connections in breast cancer patients with CRCI [8,9,10]. Dumas et al. [11] reported that decreased functional connectivity (FC) one month after chemotherapy that partially recovered to baseline at one year in the dorsal attention network and decreased FC in the default mode network (DMN) at one month and one year following chemotherapy. A previous study on the hippocampus showed that this brain region in breast cancer patients after chemotherapy has higher FC than that of healthy controls and was associated with poor subjective cognitive performance [12]. Chen et al. [9] found older breast cancer patients have increased amplitude of low frequency fluctuation (ALFF) of bilateral subcallosal gyri and right anterior cingulate gyrus after chemotherapy, and decreased fractional ALFF (fALFF) in the left precuneus, while there was no significant difference in regional homogeneity (ReHo) analysis. However, most of these studies are carried out separately according to the brain functional connectivity or regional activity, and the result is based on the comparison of the population level. Given the multimodal characteristics of brain function, it will be a challenge to identify the distinguishing features and apply them to the individual classification of CRCI, particularly for the survivors at the period of SCC.

To solve these problems, the classification models combining rs-fMRI with machine learning has been applied to the early accurate diagnosis of CRCI. Kesler et al. [13] constructed the support vector machine (SVM) with patterns of DMN connectivity, and distinguished CRCI from both survivors without chemotherapy and HCs with 90–91% accuracy, while the effect of other parameters of rs-fMRI functional connection and activity on breast cancer patients after chemotherapy was ignored. Furthermore, there were insufficient concerns for breast cancer survivors who were still experiencing chemotherapy-related SCC. Previous studies have suggested that the combination of multi-level rs-fMRI features could improve the disease diagnosis classification performance [14,15].

Based-on these, in this present study, we assumed that we can use the method of machine learning to construct the SVM models through a variety of rs-fMRI functional parameters, including FC, ALFF, fALFF, ReHo, voxel-mirrored homotopic connectivity (VMHC) and degree centrality (DC), which could effectively identify breast cancer patients with chemotherapy-related SCC from breast cancer patients before chemotherapy and HCs, and these might provide a valuable message for the accurate diagnosis of the occurrence of CRCI and clinical intervention in breast cancer patients.

## 2. Materials and Methods

### 2.1. Participants

All breast cancer patients were recruited in the First Affiliated Hospital of Nanchang University from July 2018 to May 2020. Patient inclusion criteria include the following: (1) female patients with primary invasive breast cancer confirmed by postoperative pathologically, and with no brain metastasis confirmed by MRI; (2) right-handed; (3) aged 35 to 64 years old; (4) appearance of subjective cognitive complaints that are not related to the disease and corresponding treatment after chemotherapy in post-operative chemotherapy. SCC are defined as self-reports of memory, execution, attention, processing speed and other cognitive impairment. Exclusion criteria include the following: (1) non-primary invasive breast cancer; (2) any history of brain trauma, tumor, nerve demyelination, mental illness, or alcohol and drug abuse; (3) any contraindications to MRI scanning (vertigo, claustrophobia, etc.). According to the above criteria, a total of forty-six breast cancer patients with subjective cognitive complaints after postoperative chemotherapy (SCC group) who met above inclusion criteria were recruited. Additionally, fifty-one breast cancer patients before postoperative chemotherapy (BC group) who met the above first three inclusion criteria and all exclusion criteria were recruited as control group. Meanwhile, thirty-four age- and sex-matched healthy controls (HC group) were volunteered from the local community, which according with above exclusion criteria.

This study was approved by the Medical Research Ethics Committee and the Institutional Review Board of the First Affiliated Hospital of Nanchang University (Protocol number 2018039 from 1 July 2018) and was performed in accordance with the Declaration of Helsinki. Each subject signed a written informed consent before participating in this study.

### 2.2. Cognitive Assessment

The subjective and objective cognitive performance of breast cancer patients was measured by using neuropsychological cognitive assessments, and it was completed within the same day of MRI scanning. The subjective cognitive complaints were estimated by the questionnaire of Functional Assessment of Cancer Therapy-Cognitive Function (FACT-Cog) [16] version 3. There are four subdomains to evaluate the perceived cognitive impairment, impact of perceived cognitive impairments on quality of life, comments from others, and perceived cognitive abilities. Additionally, objective cognition would be measured from different cognitive subdomains by the following six neurophysiological tests: (1) Montreal Cognitive Assessment Beijing version [17]; (2) Trail making test [18]: it consists of parts A and B; (3) Stroop color-word test (CWT) [19]: there are three parts in this test, including word test, color test and color-word test; (4) The Chinese version of auditory verbal learning test (AVLT) [20]: it included immediate and delayed recall; (5) Symbol digital modalities test [21]; (6) Clock drawing test [22].

Moreover, the Beck Depression Inventory [23] was used to assess the severity of depression in patients, the higher scores the more severe the symptoms.

### 2.3. MRI Data Acquisition

All MRI data were collected on a 3.0-Tesla Siemens Trio TIM Scanner (Siemens Medical Solutions, Erlangen, Germany) using an eight-channel head coil at the Radiology Department of the First Affiliated Hospital of Nanchang University. To minimize head motion and discomfort, all participants laid supine with their head fixed using foam pads in the scanner. During the MRI data acquisition, the participants were required to keep their eyes closed without thinking anything specifically or falling into sleep.

The rs-fMRI gradient-echo echo-planar imaging sequences were collected as follows: repetition time = 2000 ms, echo time = 30 ms, flip angle = 90°, matrix size = 64 × 64, field of view = 220 × 220 mm^2^, voxel size = 3.0 × 3.0 × 4.0 mm^3^, inter-slice gap = 0 mm, slice thickness = 4 mm, 30 axial slices, and 240 time points with a scan time of 8 min and 40 s. The high-resolution T1-weighted images were collected using three-dimensional magnetization-prepared rapid gradient-echo sequence with the following parameters: repetition time = 1900 ms, echo time = 2.26 ms, inversion time = 900 ms, matrix size = 240 × 256, field of view = 250 × 250 mm^2^, voxel size = 1.0 × 1.0 × 1.0 mm^3^, slice thickness = 0.5 mm, and 176 sagittal slices with a scan time of 4 min and 20 s. Each participant also underwent conventional sequences including T1-weighted, T2-weighted, T2-weighted dark fluid, and diffusion-weighted imaging sequences to analyze brain structural injury and metastasis.

### 2.4. MRI Data Preprocessing

The conventional sequence images and diffusion weighted imaging were checked by three experienced radiologists to weed out structural brain injury and metastasis before preprocessing. The rs-fMRI and high-resolution T1-weighted images were preprocessed by using a toolbox called Data Processing Assistant for Resting-State fMRI (http://rfmri.org/DPARSF (accessed on 20 November 2021)) [24] based on Statistical Parametric Mapping 12 (http://www.fil.ion.ucl.ac.uk/spm/ (accessed on 20 November 2021)) running on MATLAB 8.4.0. (Mathworks Inc., Sherborn, MA, USA) through the following seven steps: (1) removing the first ten time points for magnetization stabilization and adaptation the environment; (2) slice-timing correction; (3) realignment to corrected head motion (whose maximum head motions exceed 2.0 mm of displacement in any direction or 2 degrees in any angular motion should be excluded); (4) spatial normalization: the individual registration between functional images and high-resolution T1 data, then T1 images segmentation and functional data registration to Montreal Neurological Institute space, and resampling to 3 × 3 × 3 mm^3^ voxel; (5) smoothing by using a 6 × 6 × 6 mm^3^ full-width at half-maximum Gaussian kernel; (6) linear de-trending to remove nuisance signals and linear regression to obtain Friston 24-head motion parameters, cerebrospinal fluid and white matter signal; and (7) band-pass filtering at 0.01–0.08 Hz to control high-frequency physiological noise and low-frequency drift.

### 2.5. Feature Extraction

After data preprocessing, based on 90 brain regions of the automated anatomical labeling (AAL) atlas, we calculated the following six types of rs-fMRI measures, including FC, ALFF, fALFF, ReHo, VMHC and DC as the variables.

(1)FC can reflect the temporal correlation between neurophysiological activities in different brain regions spatially. The mean time courses of each regions of interest (ROI) were attained by averaging the rs-fMRI time series of all the voxels within the ROI. Pearson’s correlation analysis was performed on all pairs of ROIs time courses, and then the correlation coefficients were transformed into z-values with Fisher’s r-to-z transformation. Total 4005 z-transformed correlation coefficients were taken as the FCs between all pairs of ROIs.(2)ALFF and fALFF reflect the intensity of intrinsic spontaneous brain oscillatory activity. ALFF was achieved as the averaged square root in a voxel over the preceding frequency range, while fALFF was calculated as the ratio between ALFF and the average square root of the power spectrum within the entire frequency range. We calculated individual ALFF/fALFF values within each voxel and then matched the mean to the brain maps. Fisher’s r-to-z transformation was performed to obtain zALFF/zfALFF map of the whole brain. Finally, we obtained 90 zALFF/zfALFF values using the AAL atlas.(3)ReHo as a local brain connectivity metric is measured to detect the regional temporal homogeneity of neural intrinsic activity. After spatial normalization, all images were then band-pass filtered as described at step 7 and smoothing would be conducted. We calculated individual ReHo values within each voxel and adjacent 26 voxels, then the mean be segmented into 90 ROIs. Fisher’s r-to-z transformation was performed to obtain zReHo map of the whole brain as well. Thus, we obtained 90 zReHo values.(4)VMHC can reflect the functional connectivity of mirrored voxels in two hemispheres. Individual VMHC values within each voxel were calculated, then segmented into 90 ROIs. After performing Fisher’s r-to-z transformation, we obtained 90 zVMHC values of brain map.(5)DC reflects the importance of this brain region by calculating the number of functional connections directly connected with this brain region in the whole brain. Each region’s DC with positive binary values of AAL atlas would be calculated and transformed to zDC values, then we obtained 90 zDC values.

### 2.6. Feature Selection

First, we performed two-sample *t*-test on each feature between the SCC and BC group, and the features that were significantly different (*p* < 0.01) between the two groups were retained. Then, we eliminated the features with high pairwise correlation to weaken the multi-collinearity. If there was a high Pearson’s correlation between two features (we set 0.65 as the correlation absolute threshold), we checked the mean absolute correlation of each variable and removed the variable with a larger mean absolute correlation. Lastly, we used the least absolute shrinkage and selection operator (LASSO) logistic method with 10-fold crossing validation using classification accuracy as cost function to select the most discriminative features for classification. After LASSO, the features whose coefficients were non-zero were remained to train a model for classifying the SCC and BC group. The features for classifying the SCC and HC group were selected as per the above identical procedures. The feature selection was performed in Python 3.8.8 with packages “scikit-learn”.

### 2.7. SVM Classification

Based on the selected features, we constructed SVM models to discriminate the three groups (SCC vs. BC, SCC vs. HC). To enhance the interpretability of the models, the linear kernel function was considered in this study. The regularization parameter *C* in the linear kernel SVM models was used to find the optimal hyperplane. The hyper-parameter *C* was left at default setting, searched by ranging from 2^−10^ to 2^10^ with 2^0.2^ as step sizes via five-fold cross-validation. The leave-one-out cross-validation method and permutation test (5000 times) were used to build the optimal SVM model. The accuracy, sensitivity and specificity of models were calculated, and the receiver operating characteristic (ROC) curve and the area under the curve (AUC) are used to evaluate the performance of the models. The weights of all selected rs-fMRI features were calculated to quantify the contribution to the SVM models. The procedures of building the linear kernel SVM models were performed in MATLAB 8.4.0. with the libsvm version 3.24 packages (https://www.csie.ntu.edu.tw/~cjlin/libsvm/) (accessed on 30 December 2021) [25].

### 2.8. Statistical Analyses

Statistical analysis was performed using SPSS 26.0 software for clinical characteristics and cognitive function assessment by independent sample *t*-test or one-way analysis of variance (ANOVA). Prior to ANOVA, clinical data were checked for normality and homogeneity of variance. If the differences were significant, they were further compared using the least significant difference test; otherwise, non-parametric tests were employed, and *p* < 0.05 was considered significant.

## 3. Results

### 3.1. Demographics and Clinical Characteristics

Two patients (all in the SCC group) with brain metastasis and six patients (four in the SCC group and two in the BC group) with excessive head motion were ruled out, and this study were recruited 40 participants in the SCC group, 49 in the BC group and 34 in the HC group finally. All patients in the SCC group treated with standard-dosage chemotherapy regimens: (1) 24 patients with docetaxel (75 mg/m^2^) + cyclophosphamide (500 mg/m^2^) + doxorubicin (50 mg/m^2^) or epirubicin (100 mg/m^2^) by intravenous infusion every 3 weeks; (2) 16 patients with docetaxel (75 mg/m^2^) + cyclophosphamide (500 mg/m^2^) + fluorouracil (500 mg/m^2^) + doxorubicin (50 mg/m^2^) or epirubicin (100 mg/m^2^) by intravenous infusion every 3 weeks. Seventeen patients in the SCC group have completed the chemotherapy treatment at the time of the MRI scan, while the rest were still undergoing treatment (6 patients completed two cycles, 6 patients completed five cycles, 5 patients completed six cycles, and 6 patients completed seven cycles). Demographic and clinically relevant characteristics and cognitive function parameters are presented in Table 1. No significant differences in age, years of education, or head motion were found among these three groups. There were no significant differences between the SCC and BC group in the Beck Depression Inventory and objective cognitive assessment scores. However, the SCC group had significantly worse subjective cognitive performance, including the FACT-Cog total scores, perceived cognitive impairment and comments from others, compared with the BC group (Table 1).

### 3.2. Feature Selection

Between the SCC and BC group, we finally obtained 38 rs-fMRI features, including 25 FC, 6 ALFF, 2 fALFF, 4 ReHo and 1 VMHC after the feature selection procedure (Figure 1 and Figure 2, Table 2). Between the SCC and HC group, we finally obtained 17 rs-fMRI features, including 7 FC, 2 ALFF, 4 fALFF, 3 ReHo and 1 DC after feature selection (Figure 3 and Table 3).

### 3.3. SVM Classification

The regularization parameter *C* in the SVM models were determined as 2^2.6^ (SCC vs. BC) and 2^0^ (SCC vs. HC), respectively, by iterative attempts using the 5-fold cross-validation scheme. With the 38 rs-fMRI features having been chosen after the above three screening procedures, after the hyper-parameter optimization and leave-one-out cross-validation, we obtained the optimal SVM model with an accuracy of 82.0% and an AUC of 0.903 (95% confidence interval: 82.2–95.6%) for discriminating the SCC from BC group (sensitivity = 82.50%, specificity = 87.76%, and the *p* values was 0.0003 after 5000 times permutation test) (Figure 4). With the 17 rs-fMRI features, we built optimal SVM model for discriminating the SCC from HC group with an accuracy of 91.9% and an AUC of 0.943 (95% confidence interval: 86.3–98.3%) (sensitivity = 92.50%, specificity = 91.18%, and the *p* values was 0.0004 after 5000 times permutation test) (Figure 4).

## 4. Discussion

In the present study, we constructed the SVM models with excellent performance that combined a variety of rs-fMRI functional parameters, including FC, ALFF, fALFF, ReHo, VMHC and DC with machine learning method, which could effectively discriminate breast cancer patients with chemotherapy-related SCC from breast cancer patients before chemotherapy and HCs, and these suggested the remarkable potentialities of rs-fMRI to identify CRCI and could provide more evidence for early clinical intervention.

### 4.1. SVM Classification

Based on multi-level rs-fMRI features, this study utilized one of the most popular machine learning classifiers SVM to discriminate breast cancer patients with chemotherapy-related SCC in breast cancer survivors with excellent classification accuracy and reliability. Kesler et al. [13] analyzed the DMN resting-state functional connectivity patterns, combined the SVM model, and discriminated chemotherapy-treated from non-chemotherapy-treated breast cancer survivors and healthy controls with high accuracy, suggesting DMN functional connectivity might be used as a neuroimaging marker for predicting CRCI. While the features type was limited to FC, local functional activity metrics were not considered. Similarly, Hosseini et al. [26] applied the multi-variate pattern analysis method to decoding the brain functional connectivity pattern during an executive-prefrontal task fMRI and built an SVM model for distinguishing chemotherapy-treated breast cancer patients. The connectivity of frontal lobe, supplementary motor area, and inferior parietal lobule brain regions contributed significantly to the model, but the classification performance was mediocre (the accuracy was about 70%). Kesler et al. [27] combined connectome-based predictive modeling [28] with rs-fMRI, built a random forest classifier model for unsupervised machine learning, and obtained three different biotypes in the breast cancer group with chemotherapy treatment. There were significant differences in cognitive impairment types and brain functional network connection patterns among patients with different biotypes. It suggested that there may be various clinical types of CRCI in breast cancer, which could provide new insights into the heterogeneity of cognitive symptoms in CRCI, but a large sample longitudinal follow-up study design is needed in the future to track individual cognitive trajectories.

In addition, brain local functional variables and white matter network characteristics were also used to classify breast cancer patients after chemotherapy. Based on rs-fMRI and generalized q-sampling imaging, Chen et al. [29] constructed multiple machine learning models separately for classification of post-chemotherapy breast cancer patients versus healthy controls. Among them, the following showed good performance, and the classification accuracies could reach more than 80%: the logistic regression characterized by ReHo, generalized fractional anisotropy and normalized quantitative anisotropy; the classification and regression tree characterized by generalized fractional anisotropy; the XGBoost classifier characterized by normalized quantitative anisotropy.

Furthermore, the effectiveness of these three classifiers was also verified by functional network and structural network constructed using graph theory analysis [30]. Among the features of topological network, the classification and regression tree and XGBoost constructed with the functional global efficiency as feature and the classification and regression tree model constructed with the structural transitivity as feature still obtained good accuracies (classification accuracies could reach more than 80%). Graph theory analysis simplified the connectivity relationships of complex brain topological networks. The use of different network metrics to quantitatively assess alterations in brain integrative and isolated functions may also provide new directions for investigating neuroimaging diagnostic markers of cognitive changes underlying CRCI in breast cancer.

In this study, combined with the multi-level rs-fMRI features included brain functional activity (FC and VMHC), local functional activity (ALFF, fALFF and ReHo) and graph theory analysis (DC), and selected by *t*-test, removal of high pairwise correlation and LASSO regression, we finally built the linear kernel SVM models with excellent classification accuracy for distinguishing breast cancer patients with chemotherapy-related SCC from breast cancer patients before chemotherapy and healthy controls.

### 4.2. Rs-fMRI Features

The features screened between the SCC and BC group covered a wide range of brain network areas, but they were mainly distributed in DMN and subcortical regions. DMN was a set of brain regions that included hubs around the anterior and posterior medial cortex, bilateral temporal lobes, as well as superior frontal and parietal cortices, which characterized by whose activity is high when the mind is not engaged in specific behavioral tasks and low during focused attention on the external environment. These regions played important roles in internal orientation cognition, cognitive control and self-reference cognitive processing [31], which closely associated with a variety of neuropsychiatric diseases [32]. For instance, Feng et al. [33] reported the significant alterations of hippocampal functional connectivity with DMN mainly in the left insula, temporal lobe and the left inferior frontal gyrus in breast cancer patients underwent chemotherapy. In the previous study [13], the difference of functional connectivity in DMN was used as the imaging features for identifying breast cancer patients receiving chemotherapy. These findings suggested that abnormal functional alterations in DMN may be one of the most effective biomarkers for distinguishing CRCI in breast cancer.

The subcortical regions involved hippocampus, amygdala and basal ganglia (caudate nucleus, putamen, pale balloon and claustrum) and other brain regions, which were associated with cognitive, emotional and social functioning, such as emotional regulation, long-term memory and spatial orientation [34,35,36,37]. Furthermore, the other brain networks involved in this mechanism, such as the frontal-parietal control network (FPCN), salience network (SN), cingular-opercular network (CON), auditory network (AN) and visual network (VN), suggested that cognitive impairment caused by chemotherapy may involve abnormal alterations in a wide range of functional networks in the whole brain.

The features screened between the SCC and HC group seldom overlapped with those between the SCC and BC group, but the engaged brain network areas were comparable. In addition to the highest proportion in DMN regions, the rs-fMRI features selected between the SCC and HC group were also mainly involved FPCN and CON related brain regions. The FPCN, also called central executive network, including prefrontal lobe, dorsolateral prefrontal lobe, dorsolateral superior frontal lobe/anterior cingulate gyrus, inferior parietal anterior lobule and anterior insular cortex, which mainly involved in executive, top-down cognitive control processes and externally oriented cognition [38]. Miao et al. [39] took anterior cingulate cortex as the seed regions, reported there were lower functional connectivity with the bilateral superior frontal gyrus and cuneus, left medial frontal gyrus and middle temporal gyrus in breast cancer patients after receiving chemotherapy, and significantly correlated with the executive function. Chen et al. [40] also demonstrated that the breast cancer survivors who had undergone chemotherapy showed significantly lower regional functional activity in prefrontal cortex regions, further presented the disorder of functional activity of FPCN brain area in breast cancer survivors.

The CON involved dorsal anterior cingulate gyrus/medial superior frontal cortex, anterior insula/frontal operculum and anterior prefrontal cortex area, was generally associated with the maintenance of cognitive setting during performance of cognitive tasks of external attention [41], and played a critical and causal role in switching between the FPCN and the DMN [42]. In a visual episodic memory task-fMRI study, Pergolizzi et al. [43] reported the increased activation of prefrontal cortex regions for the breast cancer group compared to controls for both before and after chemotherapy treatment; in addition, Chen et al. [40] showed similar findings by an rs-fMRI study, remarking that the alterations of functional activity in the CON might be one of the effective biomarkers to identify breast cancer patients after chemotherapy with cognitive impairment.

### 4.3. Comparison of Subjective and Objective Cognition between the SCC and BC Group

In this study, there was no significant difference in objective cognitive scores between the SCC and BC group, which corresponds to the results of previous studies [2,44]; meanwhile, in the subjective questionnaire, the total score of FACT-Cog in the SCC group was significantly lower than that in the BC group, and the perceived cognitive impairment and comments from others were significantly decreased in the SCC group, which indicated that the patients in the SCC group suffered more subjective cognitive impairment. Although there were no clear diagnostic criteria for FACT-Cog to confirm the decline of subjective cognitive function, according to a previous study [45], the total score of FACT-Cog decreased by 6.9–10.6 points, which could be diagnosed as subjective cognitive impairment. It suggested that the SCC group in this study was in accordance with the diagnosis of subjective cognitive impairment than the BC group at the group level.

### 4.4. Limitations

The current study has some limitations. First, the number of samples was still small, so increasing the sample size and verifying the model through external data sets would be our next research direction. Then, although the brain atlas used in this study is limited to the cerebrum and did not involve the cerebellum, the performance of the models were excellent, and other brain atlas and functional activity of the cerebellum could be considered in the future. Third, we only discussed the availability of functional features for rs-fMRI in this study. Given the multi-modal characteristics of MRI, a future study that combined with other MRI features such as gray and white matter volume and white matter fiber tract connectivity for the prediction of CRCI in breast cancer would be beneficial.

## 5. Conclusions

In this study, our findings demonstrated a valid machine learning approach using multiple rs-fMRI features that could effectively identify breast cancer patients with chemotherapy-related SCC from breast cancer patients without chemotherapy and healthy controls, providing potential neuroimaging evidence for early diagnosis of the occurrence of CRCI and clinical intervention in breast cancer patients.

## Figures and Tables

**Figure 1 jcm-11-02267-f001:**
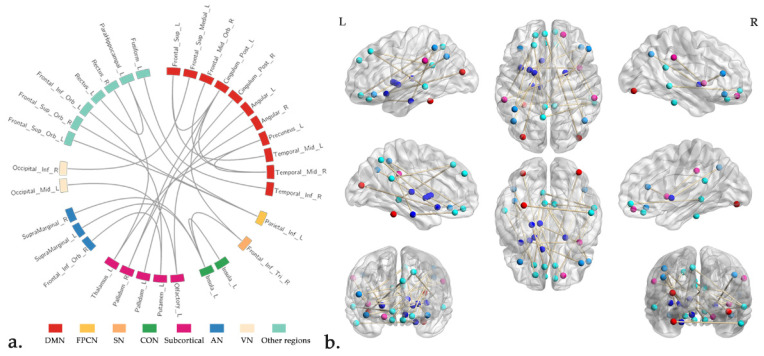
The selected FC features for discriminating the SCC from BC group. Part (**a**) showed the distribution of FC features in different brain network regions, and part (**b**) showed the spatial distribution of the FC features across the brain. Abbreviations: DMN, default mode network; FPCN, frontal-parietal control network; SN, salience network; CON, cingular-opercular network; AN, auditory network; VN, visual network; L, left; R, right.

**Figure 2 jcm-11-02267-f002:**
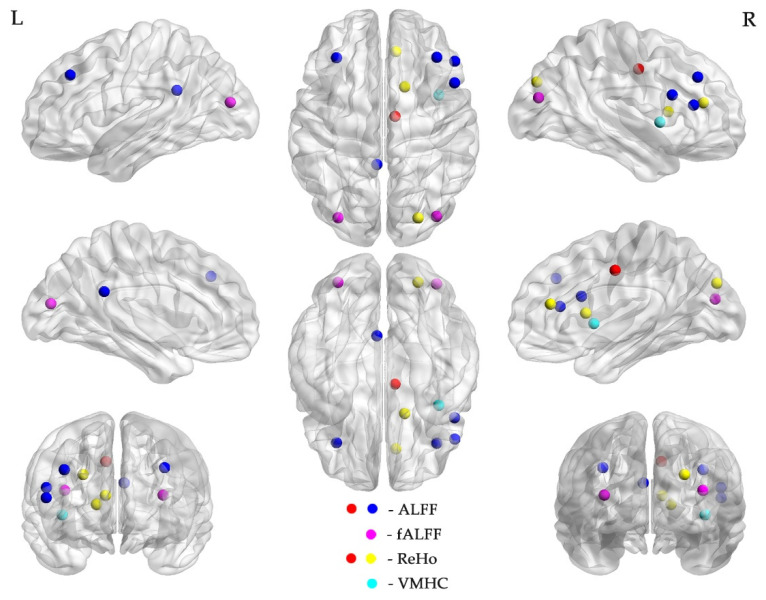
The selected ALFF, fALFF, ReHo and VMHC features for discriminating the SCC from BC group. Abbreviations: ALFF, amplitude of low frequency fluctuation; fALFF, fractional ALFF; ReHo, regional homogeneity; VMHC, voxel-mirrored homotopic connectivity; L, left; R, right.

**Figure 3 jcm-11-02267-f003:**
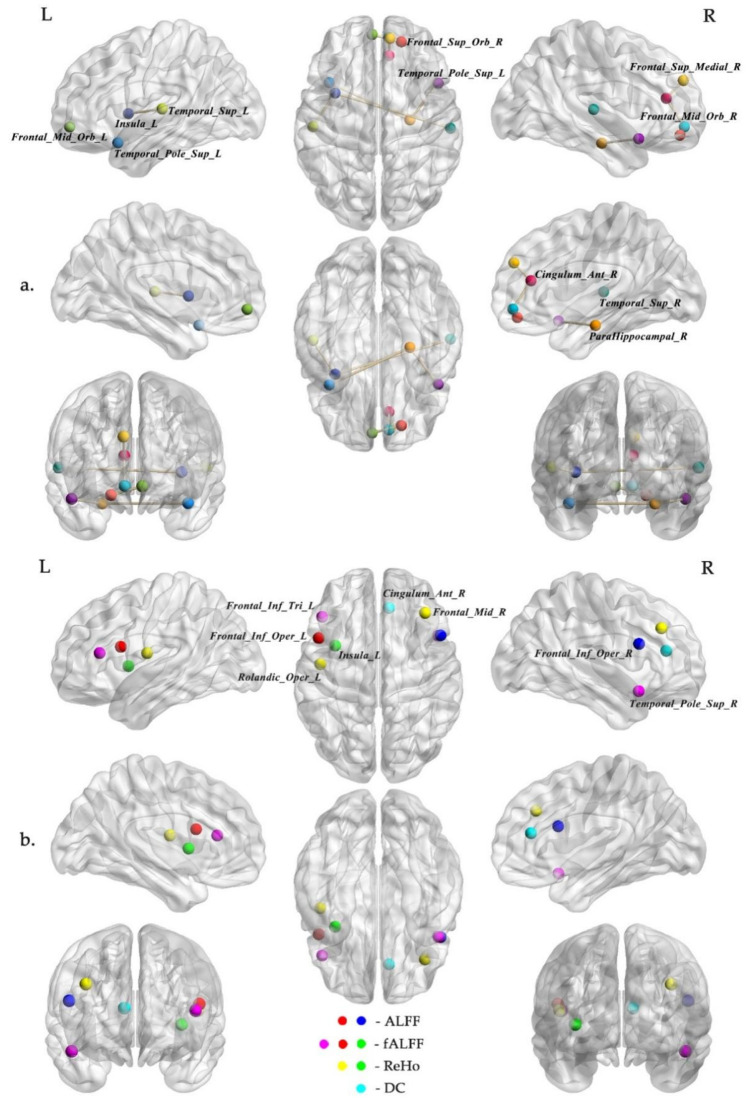
The selected rs-fMRI features set for discriminating the SCC from HC group. Part (**a**) showed the FC features, and part (**b**) showed the ALFF, fALFF, ReHo and DC features. Abbreviations: ALFF, amplitude of low frequency fluctuation; fALFF, fractional ALFF; ReHo, regional homogeneity; DC, degree centrality; L, left; R, right.

**Figure 4 jcm-11-02267-f004:**
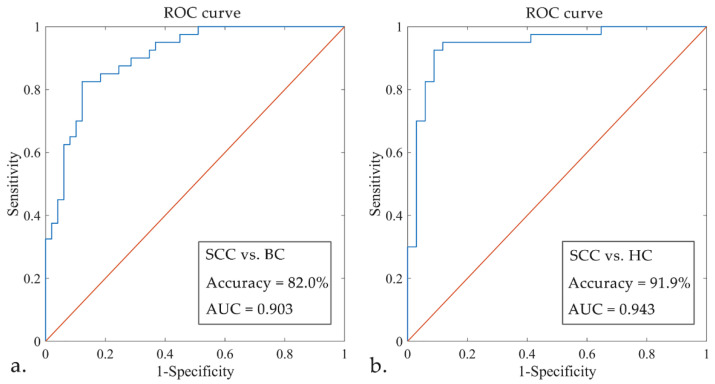
The ROC curves of SVM models. Part (**a**) showed ROC curves of SVM models for discriminating the SCC from BC group, and part (**b**) showed ROC curves of SVM models for discriminating the SCC from HC group. Abbreviations: ROC, receiver operating characteristic; AUC, area under the curve.

**Table 1 jcm-11-02267-t001:** Comparison of clinical characteristics and cognitive function among the three groups.

	SCC (N = 40)		BC (N = 49)		HC (N = 34)		*p* Values
	Mean	SD	Mean	SD	Mean	SD	
Age (years)	47.85	6.87	46.98	6.64	46.38	9.88	0.712
Education (years)	4.60	3.63	5.58	3.86	5.74	3.12	0.317
Stage of tumor (I, II, III, IV)	0, 24, 15, 1		8, 27, 11, 3				
From the start of chemotherapy (days)	205.21	272.17					
Head motion (mm)	0.11	0.05	0.10	0.04	0.10	0.05	0.585
Beck Depression Inventory scores	14.63	7.91	12.04	9.50			0.173
FACT-Cog total scores	95.89	16.44	106.94	10.95			<0.001
Perceived cognitive impairment	53.13	9.74	60.79	6.39			<0.001
Quality of life	12.23	3.02	13.31	3.40			0.120
Comments from others	14.13	2.63	15.42	1.03			0.002
Perceived cognitive abilities	16.41	3.66	17.42	3.80			0.210
Montreal Cognitive Assessment scores	20.83	4.52	20.51	4.44			0.742
Trail making test—A (s)	36.28	57.23	20.40	21.04			0.163
Trail making test—B (s)	198.10	89.21	186.66	84.47			0.638
CWT—word test (s)	31.16	18.29	28.95	11.66			0.525
CWT—color test (s)	40.22	9.19	41.09	14.49			0.766
CWT—color-word test (s)	78.69	20.69	74.57	23.06			0.425
AVLT—immediate recall	40.23	12.37	40.45	11.14			0.944
AVLT—delayed recall	9.88	3.56	9.16	3.03			0.311
Clock drawing test	19.08	6.15	20.72	7.84			0.284
Symbol digital modalities test	29.79	14.42	32.79	14.78			0.344

Abbreviations: SCC, breast cancer patients with postoperative chemotherapy-related subjective cognitive complaints; BC, breast cancer patients with postoperative non-chemotherapy; HC, healthy control; SD, standard deviation; FACT-Cog, Functional Assessment of Cancer Therapy-Cognitive Function; CWT, Stroop Color-Word Test; AVLT, Auditory Verbal Learning Test.

**Table 2 jcm-11-02267-t002:** The selected rs-fMRI features set for discriminating the SCC from BC group.

ID	Features	Brain Network	SCC (N = 40)		BC (N = 49)		*p* Values	Weight
			Mean	SD	Mean	SD		
1	Pallidum_R–Angular_R	Subcortical-DMN	0.222	0.179	0.115	0.158	0.004	5.629
2	Temporal_Mid_R–Cingulum_Post_L	DMN-DMN	0.467	0.271	0.602	0.189	0.007	−3.573
3	Insula_R–Frontal_Inf_Tri_R	CON-SN	0.439	0.233	0.560	0.188	0.008	−2.735
4	Pallidum_L–Precuneus_L	Subcortical-DMN	0.295	0.191	0.295	0.191	0.006	2.221
5	Pallidum_L–Cingulum_Post_R	Subcortical-DMN	0.264	0.166	0.15	0.158	0.001	2.159
6	Cingulum_Post_L–Frontal_Inf_Tri_R	DMN-SN	0.154	0.227	0.023	0.228	0.008	2.144
7	Putamen_L–Frontal_Inf_Orb_L	Subcortical-Other regions	0.337	0.208	0.44	0.143	0.007	−1.888
8	Parietal_Inf_L–ParaHippocampal_L	FPCN-Other regions	0.156	0.229	0.287	0.167	0.002	−1.867
9	Thalamus_L–Rectus_R	Subcortical-Other regions	0.125	0.203	0.229	0.160	0.008	−1.433
10	Angular_L–Olfactory_L	DMN-Subcortical	0.145	0.210	0.281	0.224	0.004	−1.284
11	Temporal_Inf_R–Fusiform_L	DMN-Other regions	0.709	0.296	0.864	0.217	0.005	−1.150
12	Insula_R–Insula_L	CON-CON	1.134	0.215	1.261	0.232	0.009	−1.063
13	ParaHippocampal_L–Rectus_L	Other regions-Other regions	0.401	0.264	0.543	0.197	0.005	−1.027
14	Parietal_Inf_L–Frontal_Sup_Orb_R	FPCN-Other regions	0.293	0.191	0.410	0.162	0.002	−0.993
15	Temporal_Mid_R–Precuneus_L	DMN-DMN	0.557	0.282	0.688	0.183	<0.010	−0.918
16	SupraMarginal_R–Insula_R	AN-CON	0.593	0.241	0.744	0.219	0.003	−0.880
17	Cingulum_Post_L–Frontal_Sup_L	DMN-DMN	0.591	0.202	0.712	0.224	<0.010	−0.690
18	Parietal_Inf_L–Frontal_Sup_Orb_L	FPCN-Other regions	0.352	0.202	0.463	0.194	0.009	0.678
19	Temporal_Mid_R–Frontal_Mid_Orb_R	DMN-DMN	0.447	0.229	0.588	0.184	0.002	−0.526
20	Pallidum_R–Cingulum_Post_L	Subcortical-DMN	0.245	0.184	0.144	0.159	0.007	0.481
21	SupraMarginal_L–Insula_R	AN-CON	0.521	0.246	0.66	0.209	0.005	0.377
22	Occipital_Mid_L–Olfactory_L	VN-Subcortical	0.301	0.256	0.439	0.187	0.004	−0.342
23	Putamen_L–Frontal_Inf_Orb_R	Subcortical-AN	0.302	0.194	0.409	0.154	0.005	−0.337
24	Temporal_Mid_L–Frontal_Sup_L	DMN-DMN	0.637	0.282	0.791	0.216	0.005	0.243
25	Occipital_Inf_R–Frontal_Sup_Medial_L	VN-DMN	0.332	0.280	0.482	0.192	0.004	0.151
26	zALFF of Frontal_Mid_R	FPCN	−0.052	0.149	0.048	0.166	0.004	−1.345
27	zALFF of Frontal_Mid_L	FPCN	−0.022	0.170	0.079	0.185	0.009	1.064
28	zALFF of Cingulum_Mid_R	SN	−0.081	0.169	0.019	0.178	0.008	−0.965
29	zALFF of Frontal_Inf_Oper_R	FPCN	−0.209	0.123	−0.116	0.132	0.001	−0.781
30	zALFF of Frontal_Inf_Tri_R	SN	−0.242	0.118	−0.156	0.115	0.001	0.265
31	zALFF of Cingulum_Post_L	DMN	0.406	0.369	0.664	0.461	0.005	0.260
32	zfALFF of Occipital_Mid_L	VN	0.894	0.371	0.69	0.316	0.006	3.148
33	zfALFF of Occipital_Mid_R	VN	0.927	0.367	0.717	0.376	0.009	0.389
34	zReHo of Occipital_Sup_R	VN	0.586	0.300	0.412	0.311	0.009	−2.314
35	zReHo of Caudate_R	Subcortical	−0.307	0.239	−0.109	0.253	<0.001	−1.966
36	zReHo of Cingulum_Mid_R	SN	0.227	0.150	0.342	0.156	0.001	−1.272
37	zReHo of Cingulum_Ant_R	DMN	−0.004	0.188	0.110	0.211	<0.010	−1.158
38	zVMHC of Insula_R	CON	0.414	0.122	0.485	0.130	<0.010	−0.542

Abbreviations: SCC, breast cancer patients with postoperative chemotherapy-related subjective cognitive complaints; BC, breast cancer patients with postoperative non-chemotherapy; SD, standard deviation; DMN, default mode network; FPCN, frontal-parietal control network; SN, salience network; CON, cingular-opercular network; AN, auditory network; VN, visual network.

**Table 3 jcm-11-02267-t003:** The selected rs-fMRI features set for discriminating the SCC from HC group.

ID	Features	Brain Network	SCC (N = 40)		HC (N = 34)		*p* Values	Weight
			Mean	SD	Mean	SD		
1	Cingulum_Ant_R–Frontal_Mid_Orb_R	DMN–DMN	0.607	0.260	0.783	0.263	0.005	−3.121
2	Temporal_Sup_L–Insula_L	AN–CON	0.822	0.202	0.650	0.246	0.001	2.750
3	Temporal_Pole_Sup_R–ParaHippocampal_R	DMN–Other regions	0.653	0.215	0.491	0.224	0.002	2.677
4	Cingulum_Ant_R–Frontal_Sup_Medial_R	DMN–DMN	0.756	0.171	0.922	0.203	<0.001	−2.285
5	Temporal_Sup_R–Insula_L	VAN–CON	0.604	0.211	0.457	0.253	0.008	1.639
6	Frontal_Mid_Orb_L–Frontal_Sup_Orb_R	DMN–Other regions	0.852	0.326	0.774	0.303	0.008	1.233
7	Temporal_Pole_Sup_L–ParaHippocampal_R	DMN–Other regions	0.606	0.254	0.430	0.248	0.004	0.004
8	zALFF of Frontal_Inf_Oper_R	FPCN	−0.209	0.123	0.080	0.189	0.001	−3.947
9	zALFF of Frontal_Inf_Oper_L	FPCN	−0.162	0.129	0.010	0.223	<0.001	0.633
10	zfALFF of Frontal_Inf_Tri_L	FPCN	0.166	0.223	0.327	0.241	0.004	−3.043
11	zfALFF of Insula_L	CON	0.220	0.232	0.383	0.264	0.006	−1.225
12	zfALFF of Temporal_Pole_Sup_R	DMN	0.471	0.236	−0.314	0.270	0.009	1.139
13	zfALFF of Frontal_Inf_Oper_L	FPCN	0.075	0.367	0.360	0.456	0.004	−0.546
14	zReHo of Frontal_Mid_R	FPCN	−0.446	0.250	−0.262	0.196	0.001	−3.864
15	zReHo of Insula_L	CON	−0.401	0.172	−0.279	0.179	0.004	−1.580
16	zReHo of Rolandic_Oper_L	AN	0.369	0.182	0.528	0.188	<0.001	−0.434
17	zDC of Cingulum_Ant_R	DMN	−0.527	0.110	−0.418	0.148	0.001	−3.346

Abbreviations: SCC, breast cancer patients with postoperative chemotherapy-related subjective cognitive complaints; BC, breast cancer patients with postoperative non-chemotherapy; HC, healthy control; SD, standard deviation; DMN, default mode network; AN, auditory network; CON, cingular-opercular network; VAN, ventral-attention network; FPCN, frontal-parietal control network.

## Data Availability

Not applicable.

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
