# Peer review of "Classification of Chemotherapy-Related Subjective Cognitive Complaints in Breast Cancer Using Brain Functional Connectivity and Activity: A Machine Learning Analysis"

_jcm, 2022, doi:10.3390/jcm11082267_

Round 1

Reviewer 1 Report

The authors present a manuscript on machine learning analyzing brain functionality in breast cancer students undergoing chemotherapy.

Generally, the manuscript is well written, the message is clear. The methods are sound and the conclusion warranted by the data. The number of patients included is on the lower edge when considering machine learning approaches.

My major concerns are:

  • I am missing the duration and dose as well as type of chemotherapy used. Assumingly, the different tumor stages as well as the age, weight, etc. of the patients gives the need of using different chemotherapy regimes and dosages. The authors did not include this information.
  • Again with chemotherapy, it is not clear if the pateitns udnerwent chemo while the scanning took place or if they already finished it and if yes, for how long

Minor points:

Introduction: Are there any studies conducted that the brain recovers after chemotherapy? e.g. a normalization after 10 years or similar?

Materials and Methods: How many breast cancer patients did you have overall to end up with 46 (i.e. how many percent did you remove?)

Reviewer 2 Report

This article is shown the classification of resting-state functional MRI features for chemotherapy-related subjective cognitive complaints in breast cancer or breast cancer or healthy control.  This is an exciting and well-written paper. I have the following minor comments:

1) Were the rs-fMRI features between BC and HC groups the same?  It is needed to describe whether are different features between BC and HC in the Result section. 

2) Is it correlated between different rs-fMRI features (SCC vs. BC) and FACT-Cog scores?
